molecular biology, biomaterials, environmental science

ocean acidification, scleractinian corals, *Stylophora pistillata*, primary polyps, *Symbiodinium*

**Author for correspondence:**
Tali Mass
e-mail: tmass@univ.haifa.ac.il

# Combined responses of primary coral polyps and their algal endosymbionts to decreasing seawater pH

Federica Scucchia[1,2], Assaf Malik[1], Paul Zaslansky[3], Hollie M. Putnam[4] and Tali Mass[1,5]

[1]Department of Marine Biology, The Leon H. Charney School of Marine Sciences, University of Haifa, Haifa 3498838, Israel
[2]The Interuniversity Institute of Marine Sciences, Eilat 88103, Israel
[3]Department for Operative and Preventive Dentistry, Charité—Center for Dental and Craniofacial Sciences, Universitätsmedizin Berlin, Berlin 14197, Germany
[4]Department of Biological Sciences, University of Rhode Island, Kingston, RI 02881, USA
[5]Morris Kahn Marine Research Station, The Leon H. Charney School of Marine Sciences, University of Haifa, Sdot Yam, Israel

  FS, 0000-0001-7785-4758; HMP, 0000-0003-2322-3269; TM, 0000-0002-7298-290X

With coral reefs declining globally, resilience of these ecosystems hinges on successful coral recruitment. However, knowledge of the acclimatory and/or adaptive potential in response to environmental challenges such as ocean acidification (OA) in earliest life stages is limited. Our combination of physiological measurements, microscopy, computed tomography techniques and gene expression analysis allowed us to thoroughly elucidate the mechanisms underlying the response of early-life stages of corals, together with their algal partners, to the projected decline in oceanic pH. We observed extensive physiological, morphological and transcriptional changes in surviving recruits, and the transition to a less-skeleton/more-tissue phenotype. We found that decreased pH conditions stimulate photosynthesis and endosymbiont growth, and gene expression potentially linked to photosynthates translocation. Our unique holistic study discloses the previously unseen intricate net of interacting mechanisms that regulate the performance of these organisms in response to OA.

## 1. Introduction

Stony corals secrete calcium carbonate skeletons which create three-dimensional reef frameworks that support the most productive and biologically diverse marine ecosystems on Earth. Corals have a biphasic life cycle with swimming planktonic larvae and sessile adults [1]. Successful settlement of larvae and their subsequent survival (known as 'recruitment') is a key element for coral reef resilience and recovery following major disturbances [2].

Young coral settlers are highly vulnerable to mortality resulting from environmental changes, such as rising seawater temperature and ocean acidification (OA) associated with global climate change [3]. OA is expected to negatively affect mineral formation of marine calcifiers, such as corals, by making it more energetically difficult for these organisms to deposit their skeletons [4]. This circumstance has led to numerous studies on the effects of OA on coral calcification, that is predicted to substantially decline over the course of this century [5]. However, great uncertainty still remains about the extent of the threat of OA to coral persistence. This is partly due to the high variability of the coral response to OA and to the persistence of coral communities that thrive in naturally low pH environments [6,7], which has shed light on the potential for coral acclimation and adaptation to the projected acidification scenarios.

Most of the studies conducted on primary polyps focused primarily on changes occurring in the skeletal development, which has been shown to be significantly delayed, with the potential for substantial deformities with decreases in seawater pH [8]. Studies on the functioning of the coral–algal symbiosis under OA are scarce, especially related to coral early-life stages. Coral symbiotic algae, which belong to the family Symbiodiniaceae, have a major role in the coral holobiont physiology and nutrition [9]. These dinoflagellates supply the coral host with photosynthetic products, thereby supporting the host metabolism, growth and reproduction [9].

A comprehensive understanding of how the earliest stages of coral, together with their algal partners, will respond to OA is critically needed to assess the capacity of these organisms to adapt and/or acclimatize to a future acidified ocean. To this end, we performed a controlled laboratory study by culturing larvae and primary polyps (metamorphosed and settled larvae) of the coral *Stylophora pistillata* under reduced pH values (high $CO_2$ concentrations) predicted by the end of the century, specifically at pH 8.2, pH 7.8 and pH 7.6 [10]. Here, we investigate larval ability to survive and successfully recruit under OA conditions, and we examine key physiological processes and skeletal characteristics influenced by changes in seawater pH, at the level of both phenotype and gene expression. Our findings reveal that the response of young corals to OA involves an intricate network of interrelated changes in the coral holobiont.

## 2. Material and methods

Coral larvae were collected from 20 randomly selected adult colonies of the stony coral *S. pistillata* on the reef adjacent to the Interuniversity Institute of Marine Sciences (IUI, 29°30′06.0″ N 34°54′58.3″ E) in the Gulf of Eilat (Israel), under a special permit from the Israeli Natural Parks Authority. All larvae were pooled together (approx. 50 larvae from each parental colony) and were transported to a controlled environment aquarium system at the Leon H. Charney School of Marine Science at the University of Haifa.

The carbonate chemistry of seawater was manipulated in the experimental aquariums by injecting $CO_2$ to reduce the ambient pH (pH 8.2) and obtain the target values of pH 7.8 and pH 7.6.

After 9 days of experimental treatment exposure, the numbers of settled primary polyps, still swimming larvae (not settled) and dead larvae (larvae dissolved during the experiment) were recorded. The primary polyps were gently removed from the chambers and they were collected for subsequent physiological, skeletal and molecular analysis (for full details, see electronic supplementary material, Methods).

## 3. Results

### (a) Larval survival and settlement

Following the 9 days experimental exposure to decreased pH conditions, we observed a reduction in the percentage of settled larvae from 82% at the control pH 8.2, to 64% at pH 7.8, and to 46% at pH 7.6, which resulted to be significantly lower than the control (Fisher's exact test, $p < 0.0001$) (electronic supplementary material, figure S2). At the intermediate pH, we detected a significantly higher percentage of larvae that were still at the planktonic or swimming stage relative to pH 8.2 (Fisher's exact test, $p < 0.0001$), while at the lowest pH, we found significantly higher larval mortality compared to the control (Fisher's exact test, $p < 0.0001$).

### (b) Physiological parameters of primary polyps and algal endosymbionts

We then explored the physiological changes that allowed the surviving individuals to successfully settle and develop into a primary polyp. Measurements of the primary polyps dark-respiration rates, indicative of the coral metabolic rates, were significantly reduced at the lowest pH condition, with an average decrease of 64 and 69% at pH 7.6 relative to pH 8.2 and pH 7.8, respectively (one-way ANOVA, $N = 6$, $F_{2,15} = 5.927$, $p = 0.01$) (electronic supplementary material, figure S3A). In contrast with the reduction in respiration rate, coral host protein concentration, used as a proxy for tissue biomass, significantly increased in both pH 7.8 and pH 7.6 compared to the control pH 8.2 (electronic supplementary material, figure S3B) (one-way ANOVA, $N = 3$, $F_{2,24} = 24.99$, $p < 0.0001$).

We then explored changes in the algal physiological attributes, and we found a significantly higher number of endosymbiotic algae per polyp surface area at both pH 7.8 and pH 7.6 relative to pH 8.2 (electronic supplementary material, figure S3C) (one-way ANOVA, $N = 3$, $F_{2,24} = 99.25$, $p < 0.05$), pointing to an enhancement of algal growth with decreasing pH. The number of symbiont cells per host protein did not show instead any significant change (electronic supplementary material, figure S3D). Furthermore, chlorophyll $a$ concentration was higher in the polyps at the lowest pH (one-way ANOVA, $N = 3$, $F_{2,24} = 73.41$, $p < 0.0001$) (electronic supplementary material, figure S3E), even though its concentration within a single algal cell did not significantly vary relative to the control pH (electronic supplementary material, figure S3F), providing further evidence of the augmented algal density at pH 7.6. In addition, higher chlorophyll auto-fluorescence was detected in the live primary polyps at the lowest pH (one-way ANOVA, $F_{2,6} = 73.38$, $p > 0.0001$) (electronic supplementary material, figure S4A,B), matching the chlorophyll $a$ concentration results (electronic supplementary material, figure S3E).

### (c) Endosymbiont photosynthetic traits

The observed changes in the algal physiological properties are indicative of variations in photosynthetic traits. Indeed, we observed an increase of photochemical efficiency and activity in the endosymbiotic algae at pH 7.6. The maximal quantum yield ($F_v/F_m$) of photosystem II (PSII) was significantly higher in the primary polyps at pH 7.6 relative to the other two pH (electronic supplementary material, figure S4C) (Kruskal–Wallis test, $H = 7.199$, $p = 0.027$). In addition, although the measurements of the initial slope ($\alpha$) and of the minimal photoinhibition point ($E_k$) showed no significant change (electronic supplementary material, table S5), the maximum values of the relative electron transport rate (rETR), were significantly elevated in the polyps at pH 7.6 relative to the control (electronic supplementary material, figure S4D) (Kruskal–Wallis test, $H = 8.025$, $p = 0.018$). Taken together, these results show that the observed increase of algal density under acidification conditions is paralleled by the enhancement of photosynthetic efficiency and activity.

We also measured the maximum values of the non-photochemical quenching (NPQ) and found no significant change among pH treatments (electronic supplementary material, table S5).

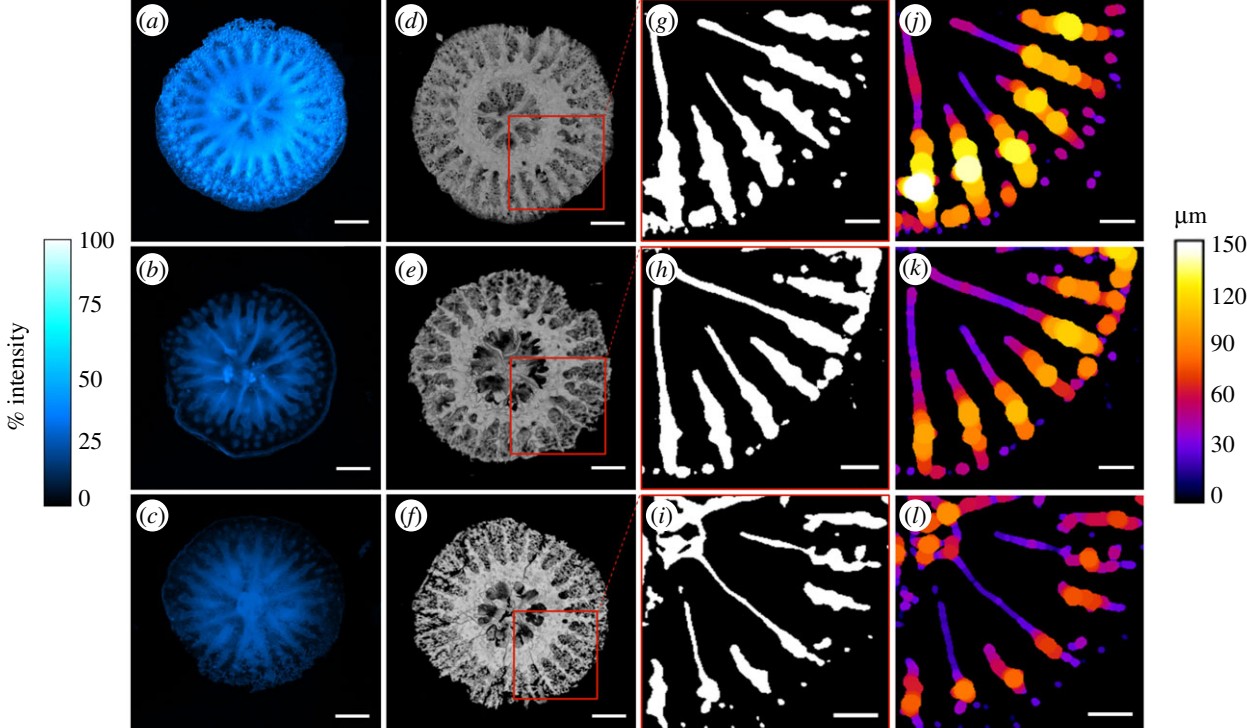

**Figure 1.** Changes in skeletal growth patterns in coral primary polyps with decreasing pH. (*a–c*) Microscopy images showing the Calcein Blue fluorescence (expressed as percentage intensity) in the skeleton of the primary polyps at (*a*) pH 8.2, (*b*) pH 7.8 and (*c*) pH 7.6. Magnification: 4×. Scale bar: 400 μm. (*d–f*) Micro-CT images showing top views of three-dimensional reconstructions of the primary polyps' skeleton at (*d*) pH 8.2, (*e*) pH 7.8 and (*f*) pH 7.6. Scale bar: 400 μm. (*g–i*) Slices of the three-dimensional reconstructions showing thresholded two-dimensional slices of the base of the septa at (*g*) pH 8.2, (*h*) pH 7.8 and (*i*) pH 7.6. Scale bar: 200 μm. (*j–l*) Skeletal thickness distribution (μm) along the vertical axis of the septa of the primary polyps at (*j*) pH 8.2, (*k*) pH 7.8 and (*l*) pH 7.6. Scale bar: 200 μm. (Online version in colour.)

## (d) Macro-scale skeletal growth patterns of primary polyps

The existence of a linkage between tissue biomass and growth [11], and our observation of increased host tissue at reduced pH, prompted us to explore changes in the primary polyps skeletal characteristics. By employing the fluorescent dye calcein [12], we visualized patterns of incorporation of divalent ions (such as $Ca^{2+}$) into the live primary polyps and subsequently into the skeleton. The *in vivo* imaging shows that in all three pH conditions, the highest calcein fluorescence intensity is found in the calyx area (see electronic supplementary material, figure S6 for additional details) and in the skeletal septa (electronic supplementary material, figure S5A). In corals, ions and other particles are ingested from seawater into the mouth cavity and are then transported to the site of calcification [13]. Our observations indicate that in all three pH conditions, corals are actively up-taking $Ca^{2+}$ from the mouth, and that these ions are eventually incorporated into the skeleton. Measurements of the calcein fluorescence intensity of the primary polyps skeleton ($N = 3$ polyps per pH condition, only 1 polyp per pH is shown in electronic supplementary material, figure S5A) show a significantly lower calcein fluorescence at pH 7.8 and 7.6 compared to pH 8.2 (electronic supplementary material, figure S5B) (one-way ANOVA, $F_{2,6} = 213.5$, $p < 0.0001$). This indicates that a lower amount of $Ca^{2+}$ was incorporated into the skeleton, thus pointing to a lower skeletal development at reduced pH conditions. Patterns of calcein incorporation correspond to the skeletal morphology imaged using micro-CT. Slices of the 3D reconstructions of the polyps skeleton (figure 1*d–f*) show a progressive reduction in the thickness of the skeletal septa as

the pH decreases (figure 1*g–l*). Such reduction, that was significant at both pH 7.8 and 7.6 compared to the control (one-way ANOVA, $F_{2,15} = 24.32$, $p > 0.001$) (electronic supplementary material, figure S5C), indicates that at reduced pH, a lower amount of $CaCO_3$ was deposited by the coral.

## (e) Skeletal micro-morphological features

Further changes in skeletal growth patterns were additionally detected at a micro-scale. Measurements of the calyx area (figure 2*a–c*, cyan overlays) of the polyps imaged by scanning electron microscopy indicate that there was no significant difference between calyxes among the three pH conditions (figure 2*m*). Differently from the calyxes, the crown areas shown in figure 2*a–c* (yellow overlays), corresponding to the forming coenosteum (see electronic supplementary material, figure S6), were significantly smaller in the primary polyps at pH 7.8 and 7.6 relative to pH 8.2 (one-way ANOVA, $N = 6$, $F_{2,15} = 8.17$, $p < 0.01$). At higher magnification, the thickness of the septa (electronic supplementary material, figure S6) was significantly smaller in polyps grown at lower pH compared to the control (figure 2*d–f,n*) (one-way ANOVA, $N = 27$, $F_{2,78} = 51.92$, $p < 0.0001$). Overall, these observations show that the reduction of skeletal development detected at the macro-scale is also manifested at the microstructural level.

Moreover, the terminal portion of the skeletal spines located on the septa showed a significantly lower number of rapid accretion deposits (RADs; globular elements in figure 2*g–i*; see electronic supplementary material, figure S6) in the polyps at reduced pH, when compared with the control (figure 2*o*) (one-way ANOVA, $N = 9$, $F_{2,24} = 19.72$, $p < 0.0001$). RADs correspond to areas of the skeleton with

*Proc. R. Soc. B* **288**: 20210328

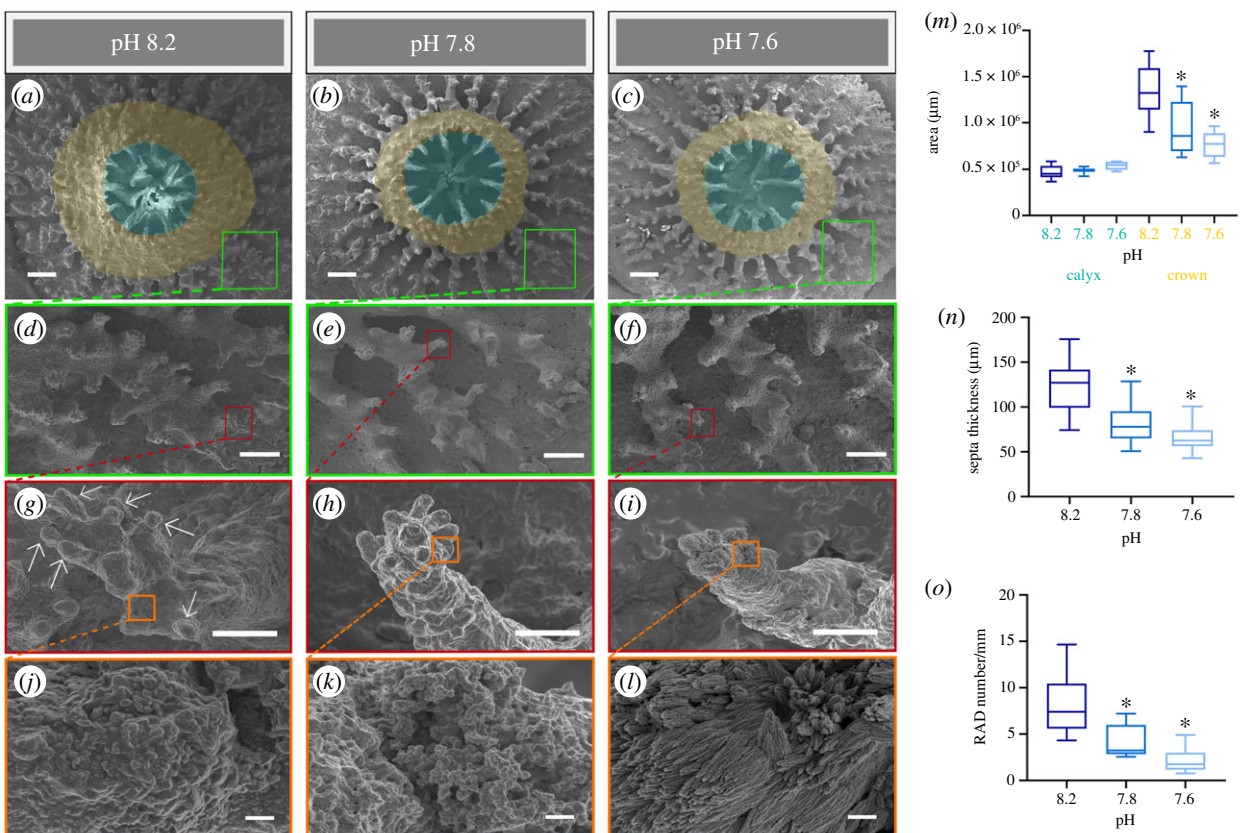

**Figure 2.** Modifications of skeletal features of coral primary polyps at different pH conditions. (*a–c*) SEM images showing the micro-morphology of the calyx (cyan areas) and crown (orange areas) of the primary polyps at (*a*) pH 8.2, (*b*) pH 7.8 and (*c*) pH 7.6. Enlargements of the skeletal (*d–f*) septa and (*g–i*) spines at (*d,g*) pH 8.2, (*e,h*) pH 7.8 and (*f,i*) pH 7.6, respectively. (*g*) White arrows indicate the RADs on the spine. (*j–l*) Insets showing the RADs surface texture at (*j*) pH 8.2, (*k*) pH 7.8 and (*l*) pH 7.6. (*m*) Size of the calyx area (cyan bar) and the crown area (orange bar) calculated in the primary polyps ($N = 6$ per pH condition). (*n*) Thickness of the septa calculated in the primary polyps at (*d*) pH 8.2, (*e*) pH 7.8 and (*f*) pH 7.6 ($N = 27$ per pH condition). (*o*) RADs number measured per units of basal area of the spines (mm) at (*g*) pH 8.2, (*h*) pH 7.8 and (*i*) pH 7.6 ($N = 9$ per pH condition). (*m–o*) Asterisks (*) indicate statistical differences ($p < 0.05$, one-way ANOVA) relative to the control pH 8.2. Scale bars: (*a–c*) 200 µm, (*d–f*) 100 µm, (*g–i*) 10 µm and (*j–l*) 1 µm.

rapid $CaCO_3$ deposition [14]. Hence, the lower abundance of RADs in the polyps at pH 7.8 and 7.6 indicates a reduced development of these regions in acidified seawater. In addition, we found prominent differences in the RADs texture. At the control pH, the texture of the RADs was smoother and more compact in comparison to the acidified conditions (figure 2*j–l*). At pH 7.6, the aragonite bundles of fibres, corresponding to the microcrystalline features on the RADs surface (figure 2*j–l*), were less compact and were characterized by granular aggregates forming needle-like structures with differing shapes and orientations (figure 2*l*) compared to the texture at both pH 8.2 and 7.8. This microstructural pattern conferred a more porous appearance to the RADs surface. The exposure to acidified seawater conditions, therefore, generates changes at multiple levels of the skeletal hierarchy, from macro- to micro-scale.

### (f) Coral host transcriptomic response to decreased pH

We examined the molecular controls underlying the observed modifications in the primary polyps physiological and skeletal characteristics. Analysis of the coral host's differentially expressed genes (DEGs) show that, compared to pH 8.2, a higher number of DEGs was found at pH 7.6 as opposed to pH 7.8 (figure 3*a*). This pattern, which reflects the logarithmic nature of the pH scale, indicates that the slope of the progression in DEGs with decreasing pH sharply steepens

between pH 7.8 and 7.6, where larger transcriptional modifications are needed to adjust to acidified seawater.

Gene Ontology (GO) enrichment and Kyoto Encyclopedia of Genes and Genomes (KEGG) pathway analyses were performed to identify enriched genes groups and molecular pathways involved in coral response to pH reduction. Figure 3*b* shows that among all pH treatments comparisons, the largest number of enriched genes groups (both up- and downregulated) was found between pH 7.6 and pH 8.2. The upregulated gene groups at the lowest pH show a significant enrichment for genes associated with the activity of voltage-gated calcium channels, with cell differentiation and developmental processes, and with genes linked to sensory perception and detection of environmental stimuli (figure 3*b*).

The downregulated genes groups at pH 7.6 include a large number of genes involved in lipid metabolism, mitochondria biosynthesis and activity, in energy production and respiration. The downregulation of these genes points to a reduced energy metabolism at acidified conditions, disclosing the molecular control behind the observed respiration rate decrease (electronic supplementary material, figure S3A).

### (g) Differential expression patterns of putative biomineralization-related genes

Looking into the molecular mechanisms underlying the observed changes in skeletal features, we evaluated the

Proc. R. Soc. B 288: 20210328

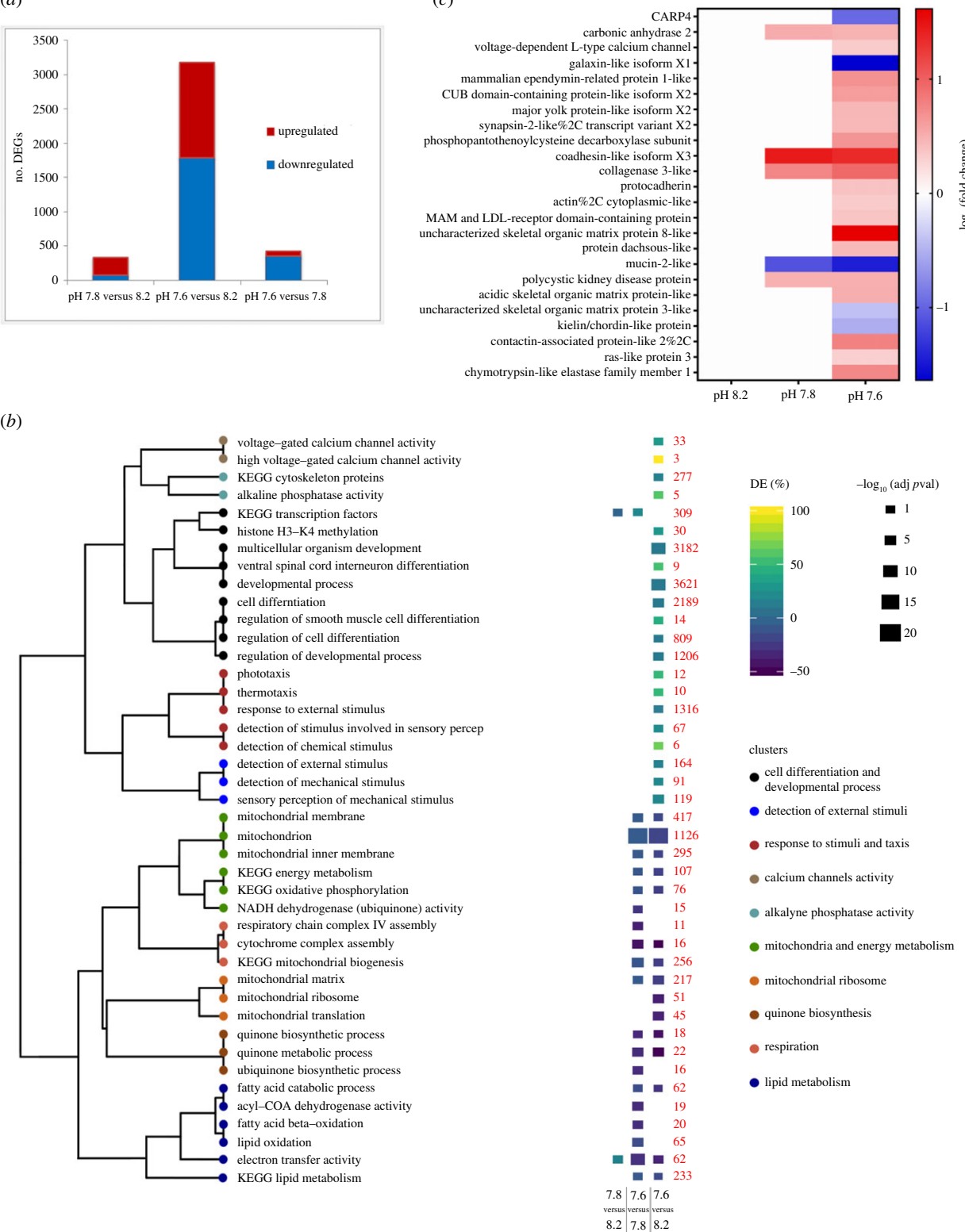

**Figure 3.** Coral DEGs, functional enrichment and biomineralization-related genes expression between experimental pH conditions. (a) Number of DEGs in coral primary polyps detected, respectively, in the pH 7.8 treatment relative to the control pH 8.2 (left bar), in the pH 7.6 treatment relative to pH 8.2 (middle bar) and in the pH 7.6 treatment relative to pH 7.8 (right bar). Red indicates the upregulated genes and blue indicates the downregulated genes within each comparison. (b) Dendrogram on the left showing enriched GO terms and KEGG pathways clustered according to the portion of identical genes shared. The heat map shows the positive (upregulated genes) or negative (downregulated genes) percentages of DEGs per GO term or KEGG pathway, detected, respectively, at pH 7.6 relative to pH 8.2 (right column) and to pH 7.8 (left column). Red numbers indicate the total number of genes within each per GO term or KEGG pathway. DE (%), percentage of DEGs. (c) Heat map showing the log2 fold change of significantly (adjusted $p$-value > 0.05) upregulated (red) and downregulated (blue) biomineralization-related genes at pH 7.8 and 7.6 when compared with pH 8.2. (Online version in colour.)

differential expression of putative coral biomineralization-related genes (electronic supplementary material, table S4). Among these 91 total genes, 5 were differentially expressed

in the polyps at pH 7.8, and 23 were differentially expressed in the polyps at pH 7.6 (figure 3c), indicating a greater response of the biomineralization molecular machinery at

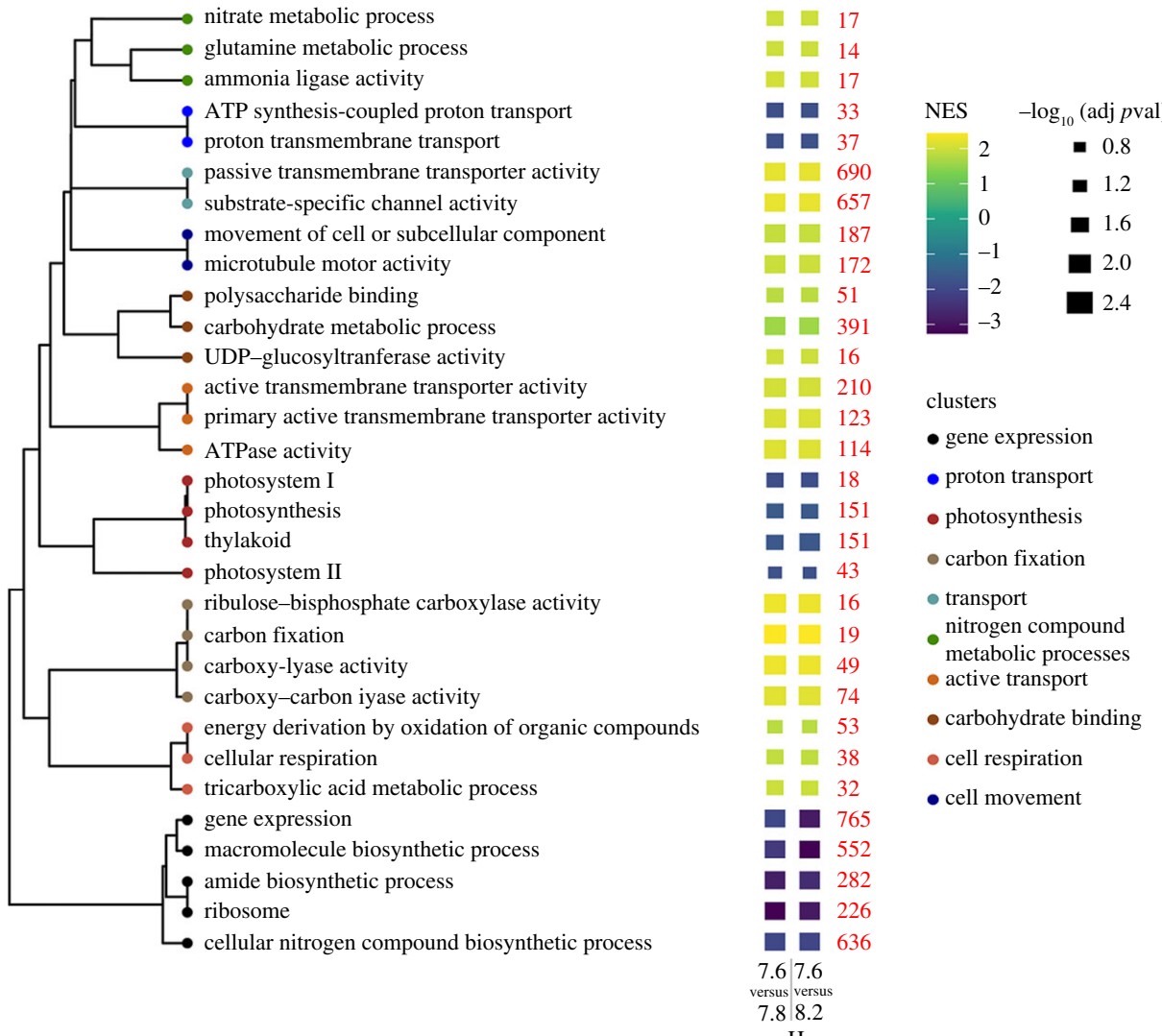

**Figure 4.** Changes in expression levels of different functional groups of genes in the coral endosymbiotic algae. Dendrogram showing enriched functional groups clustered according to the portion of identical genes shared. The heat map shows the positive (upregulated genes) or negative (downregulated genes) normalized enrichment score (obtained using the functional enrichment analysis GSEA) per each functional group, detected, respectively, at pH 7.6 compared to pH 8.2 (right bar) and to pH 7.8 (left bar). No enrichment was found in pH 7.8 compared to pH 8.2. Genes groups are shown only for cases with significant (adjusted $p$-value < 0.05) enrichment in both comparisons. Red numbers indicate the total number of genes within each functional group. NES, normalized enrichment score. (Online version in colour.)

the extreme pH. In particular at pH 7.6 relative to pH 8.2, a significantly higher number of biomineralization-related genes was upregulated, compared to the number of downregulated genes (Fisher's exact test, $p < 0.05$). Most of these upregulated genes code for proteins of the skeletal organic matrix with structural and adhesion functions. These proteins are embedded in the skeletal framework and control the growth of the aragonite crystal-like fibres within the coral skeleton [15]. Notably, we observed the downregulation of the coral acid-rich protein 4 (CARP4) gene at the lowest pH treatment (figure 3c). As member of the CARPs group, this biomineralization toolkit protein is expected to transport $Ca^{2+}$ to the calcification site that are then concentrated in centres of calcification, also known as RADs [15]. Moreover, among the upregulated genes, we detected the *S. pistillata* carbonic anhydrase 2 (STPCA-2) and the L-type calcium channel (figure 3c), which is located in the calicoblastic epithelium (cell layer overlying the coral skeleton) [16]. These two genes are involved in regulating the pH and carbonate chemistry of the calcifying medium, the confined space where skeletal deposition occurs [17,18]. In particular, the increased expression of the

L-type calcium channel, together with the enrichment of genes associated with calcium channels activity (figure 3b), indicates that the exposure to acidic seawater stimulates in the corals the transport of $Ca^{2+}$.

## (h) *Symbiodinium microadriaticum* transcriptomic response to decreased pH

In *S. pistillata* algal endosymbiont, enriched genes groups (both up- and downregulated) were only found in pH 7.6 relative to the other two pH conditions, while no enrichment was detected in pH 7.8 relative to the control (figure 4). This indicates that, similar to the coral host, wider transcriptional changes take place at the lowest pH.

The downregulated groups at pH 7.6 include genes related to photosynthesis (e.g. 'thylakoid'), gene expression (e.g. 'ribosome') and proton transport (e.g. 'proton transmembrane transport'). The upregulated groups comprise a large number of genes involved in photosynthetic carbon fixation ('ribulose-bisphosphate carboxylate activity', also known as RuBisCO, 'carboxy-lyase activity', etc.), carbohydrates binding

and transport (e.g. 'polysaccharide binding', 'substrate-specific channel activity', 'active transmembrane transporter activity'). As shown by the clustering (figure 4), terms related to carbohydrates binding and transport share a portion of common genes, suggesting that symbiont genes associated with transmembrane transport are also involved in the transport of sugars. Taken together, our observations indicate that the symbiotic algae response to reduced pH is governed by the modulation of multiple photosynthesis-related genes.

## 4. Discussion

The current study on *S. pistillata* early-life stages reveals that the exposure to predicted future OA conditions leads to a reduction of larval survival and settlement (electronic supplementary material, figure S2). Nonetheless, the more acidic seawater did not entirely inhibit larval recruitment, suggesting potential acclimatory mechanisms that allowed the corals to successfully settle and start growing. Larval recognition of the settlement substrates occurs through environmental cues, that are exogenous factors capable of inducing larval settlement [19]. Seawater acidification was shown to reduce the induction of larval settlement in response to environmental cues [20]. Based on our transcriptomic analysis, genes shown to be related to larval selection of and attachment to the substrate (i.e. sensory perception and detection of environmental stimuli-related genes) [21,22] are significantly enriched at the lowest pH (figure 3b). This indicates that the modulation of genes with a role in sensory perception may account for the adjustment capacity and survival of the corals under OA conditions, as observed in other organisms exposed to a changing environment [23].

Diminished availability of settling juveniles could inhibit the replenishment of reefs after sporadic disturbances such as storms and bleaching events [2], with the potential to compromise coral reef resilience. However, coral populations can naturally persist in acidic environments, despite having a lower recruitment efficiency in comparison to corals living at ambient pH [24]. Moreover, improved adult performance could compensate for low recruitment rates, as observed for corals living around $CO_2$ seeps [25].

In this study, larvae reared under acidified conditions that successfully metamorphosed into a primary polyp were characterized by reduced metabolic rates, as shown by the lower respiration rate (electronic supplementary material, figure S3A) and the downregulation of metabolism-related genes (figure 3b). A downward trend of energy metabolism seems a common response in young corals exposed to acidic seawater [26]. In an OA scenario, the formation of the skeleton is thought to be more energetically expensive for corals, requiring more energy to remove from the calcifying space the excess $H^+$ produced during calcification [4]. The reduction of metabolic rates observed in the primary polyps may energetically limit the rates of $H^+$ removal. This leads to a diminished ability to deposit $CaCO_3$, and indeed our multifaceted examination of the skeleton reveals that under OA primary polyps were characterized by reduced skeletal development (figures 1a–c,j–l and 2a–i,m–o; electronic supplementary material, figure S5B,C). Furthermore, our observations of the reduction in primary polyp crown areas (figure 2a–c,m) and in the number of RADs (figure 2g–i,o) show that exposure to OA conditions involves changes

within the skeleton that have never been investigated so far. In this context, the downregulation of the CARP4 gene detected at acidified conditions (figure 3c) reveals a potentially lower contribution of this protein to the formation of RADs, thereby contributing to the overall reduction in skeletal growth. In fact, this biomineralization toolkit protein was localized in *S. pistillata* skeleton in correspondence of RADs, and was suggested to form crystal binding substrates that lead to $CaCO_3$ nucleation [15].

Given the reduction of skeletal development in acidified seawater, corals must put in place alternative strategies to sustain overall growth. Indeed, the greater amount of tissue biomass covering the skeleton, detected in this study (electronic supplementary material, figure S3B), could allow coral polyps to reach larger sizes, representing a critical strategy to counterbalance the lower skeletal development. Through a lower investment of energy into skeletal growth, polyps can instead allocate more of the available energy pool to tissue biomass [27,28]. For example, in the sea urchin *Strongylocentrotus purpuratus*, another marine calcifier, protein synthesis accounted for approximately 84% of the available energy pool consumption under exposure to seawater acidification [29]. With this energy-reallocation mechanism, corals could increase their tissue biomass despite having decreased metabolic rates. Similar to our study, a higher-biomass phenotype was observed under a long-term exposure to OA in the temperate coral *Oculina patagonica* [30]. When transferred back to ambient pH conditions, these corals returned to calcify normally despite being 12 months as soft-bodied polyps in low pH conditions [30], showing that corals possess a remarkable ability to adjust and survive to future acidification conditions.

The potential ecological consequences of maintaining overall growth while reducing skeletal development are associated with reaching critical sizes for sexual maturity and assuring a high production of offspring, thereby perpetuating the species and sustaining the reef persistence [7]. The shift to a less-skeleton/more-tissue phenotype would not negatively impact the achievement of sexual maturity and gametogenesis, which appeared to develop similarly in both reduced-skeleton corals at acidified conditions and corals at ambient pH [30]. More porous skeletons with preserved rates of linear extension were observed in *S. pistillata* adults exposed to acidic conditions [31]. In addition in naturally acidic pH environments, adult corals build more porous skeletons to keep linear extension rate constant [7]. These observations indicate that, for corals, the achievement of larger sizes is indeed a critical need, and that the shift to a more porous skeleton is a common trade-off and an acclimatory strategy to cope with OA.

In acidic seawater, our results show a more porous appearance of the primary polyps' skeleton at micro-scale (figure 2l). For corals, the accretion of a more porous skeleton under OA conditions is sustained by embedding more organic matrix proteins within the skeletal fibres [31]. Indeed, our molecular examination shows an enhanced expression of organic matrix proteins at acidic conditions (figure 3c), which points to an increased incorporation of these biomineralization toolkit proteins within the skeletal pores. Previous observations report changes in the expression of organic matrix genes [26] and in the skeletal fibres arrangement [32] of early coral stages reared under OA conditions. Nevertheless, these studies focused either on the molecular or on the morphological aspects, without integrating the coral response across multiple variables.

It is noteworthy that energy investment into organic matrix synthesis is suggested to be three orders of magnitude less than the energy cost of active pumps regulating $CaCO_3$ deposition [27]. This implies that increasing the organic matrix production represents a more cost-effective way to support overall polyp growth, when less favourable conditions for skeleton building occur.

Despite the change in seawater carbonate chemistry, primary polyps still managed to favour $CaCO_3$ accretion. With our work, we show that coral recruits finely tuned the expression of calcification-related genes. In particular, our molecular analysis implies a tight control of the calcification site chemistry exerted by the coral, through the modulation of $Ca^{2+}$ transport (figure 3b,c) and of the activity of STPCA-2 (figure 3c). This latter enzyme catalyses the hydration of $CO_2$ to $HCO^{3-}$ that is subsequently delivered to the calcification site [17]. The concentrations of both $Ca^{2+}$ and $HCO^{3-}$ have a major role in the skeletal development under OA, as indicated in adults of S. pistillata and other coral species [33,34]. Furthermore, a recent study shows that genes related to $Ca^{2+}$ management and to inorganic carbon regulation (i.e. carbonic anhydrases) have a major role in the persistence of coral populations adapted to naturally low pH environments [35]. In the light of the predicted decrease of oceanic pH, the ability of corals to modulate the $Ca^{2+}$ transport and the STPCA-2 activity, especially during the delicate early-life stages, constitutes a crucial area of further research, that could disclose the susceptibility of different coral species to OA.

In symbiotic corals, metabolic requirements for growth are critically supported by the photosynthetic activity of the symbiotic algae [9]. Through the carbon fixation process, these dinoflagellates incorporate $CO_2$ into organic compounds, that are used as energy sources [9]. Growing evidence shows that enhanced $CO_2$ levels in seawater could improve the symbiont photosynthetic performance, by alleviating the carbon limitation that is experienced by symbiotic algae inside the host tissues [36,37]. The predicted increase of $[CO_2]$ in seawater [10], simulated in our study, appears to boost the activity RuBisCO (the key photosynthetic enzyme that catalyses the first major step of carbon fixation), as suggested by the increased expression of carbon fixation genes in acidic seawater (figure 4). As a result, the algal photosynthetic activity and efficiency increase (electronic supplementary material, figure S4), ultimately enhancing algal growth (electronic supplementary material, figure S3C,E).

It is noteworthy that endosymbiotic algae transfer a portion of the photosynthetically derived products, that are not ultimately used for algal growth, to the host [9]. The increased expression of genes linked to carbohydrate binding, that cluster with genes related to the activity of transmembrane transporters and channels (figure 4), suggests an enhanced transfer of photosynthates to the coral host. In corals, heterotrophic feeding requires additional energy to break down food particles and absorb nutrients, compared to sugars coming from the endosymbionts that are quickly metabolized [38]. Given the reduction of metabolic rates under OA, an increased transfer of sugars from the algal endosymbionts might constitute for corals a more cost-effective energy source, compared to heterotrophic sources. Notably, the energy derived from photosynthesis is hypothesized to have a 10–20-fold greater effect on tissue energetics compared to skeletal energetics [27]. Therefore, the augmented algal photosynthetic activity could substantially contribute to the tissue thickening response

observed in the primary polyps. In addition, photosynthetic products have been suggested to be used as precursors for skeletal organic matrix biosynthesis [39]. An increased translocation of photosynthates from S. microadriaticum would thus considerably support the enhanced synthesis of organic matrix in the corals, and it would overall increase the host ability to cope with the effects of OA [40].

Earlier works on adult corals investigating the response to OA of the algal endosymbiont focused either on transcriptomic [41] or physiological changes [42,43], and reported negative or no physiological responses in the algal symbionts with the exposure to acidic seawater. However, it must be noted that broad physiological differences exist among different species across Symbiodiniaceae [44], and that gene expression changes of the endosymbiotic algae greatly vary among coral populations in response to acidification [45]. The possibility of symbiont-mediated changes in the coral response to seawater acidification, especially at the highly critical early-life stages, urges, therefore, more in-depth future examinations.

In conclusion, we thoroughly describe the mechanisms underlying the response of S. pistillata early-life stages under OA. Our findings suggest that acclimatory mechanisms played a role in the observed coral response. However, we cannot definitively discern relative contributions of acclimation versus natural selection acting on corals genotypes, due to the differential survivorship among treatments and to genetic heterogeneity among coral larvae in the field. In any case, it is important to consider that coral recruits spawned from adults that are already experiencing the effects of decreasing pH may inherit from the parent colonies a greater tolerance to acidifying oceans [46]. These surviving individuals may be better able to cope with OA than prior generations, ultimately building coral reefs resilience through adaptive evolution (reviewed in [47]).

As summarized in electronic supplementary material, figure S9, we show that in coral primary polyps, the increased energetic demands of the calcification process under OA resulted in reduced skeletal development and higher skeletal porosity. To counterbalance these changes and support overall growth, corals enhanced the incorporation of organic matric proteins into the skeletal framework and increased the production of tissue biomass. With more available $CO_2$ dissolved in seawater, the growth and photosynthetic activity of the algal endosymbiont was greatly stimulated, leading to a potential enhancement of sugar translocation to the host.

While extrapolating laboratory-based findings to projections of the effects of OA on corals in the field is possible, we caution that controlled aquarium systems often do not simulate the dynamic environment of the reef. Moreover, other environmental factors, such as elevated seawater temperature, might act synergistically with acidification, affecting coral fitness [48]. Although extensive, our study has analysed a single coral species and at only one time point, thus limiting the power of our predictions on the acclimatory nature of the observed coral response, and on the phenotypic changes that could occur at later coral life stages under long-term acidification exposure. Thus, future efforts should focus on assessing the degree of modifications in subsequent developmental stages of various coral species under acidification conditions, allowing us to make more accurate predictions on the vulnerability of corals to future OA scenarios.

Our multidisciplinary approach reveals strong merit in investigating the response of both the host and the algal partner, as we show that the response to OA involves a wide and

intricate net of interrelated parameters in both organisms. The overall correspondence of processes across different assays and biological scales showcases the robust nature of our work, and the importance of including interdisciplinary and complementary analyses towards understanding corals' vulnerability to environmental change.

Data accessibility. All data needed to evaluate the conclusions in the paper are present in the paper and/or the electronic supplementary materials and/or are available from the Dryad Digital Repository, including the scripts employed to analyse the RNA-seq data and all the raw data of the study (i.e. raw data of physiological, morphological and photo-synthetic measurements, and of the aquariums seawater chemistry: https://doi.org/10.5061/dryad.66t1g1k27 [49]. The RNA-Seq raw data were deposited in the National Center for Biotechnology Information (www.ncbi.nlm.nih.gov/bioproject/PRJNA640748).

Authors' contributions. F.S.: conceptualization, data curation, formal analysis, investigation, methodology, project administration, writing—original draft, writing—review and editing; A.M.: data curation, methodology, software, visualization; P.Z.: data curation, methodology, resources, software, visualization; H.M.P.: conceptualization, data curation, funding acquisition, investigation, methodology, project administration, supervision, validation, writing—original draft, writing—review and editing; T.M.: conceptualization, data curation, funding acquisition, investigation, project administration, resources, supervision, validation, writing—original draft, writing—review and editing.

All authors gave final approval for publication and agreed to be held accountable for the work performed therein.

Competing interests. The authors declare no competing interests.

Funding. This project has received funding from the Israeli Binational Science Foundation (BSF 2016321 to H.M.P. and T.M.), the Israel Science Foundation (312/15 to T.M.) and from the European Research Council (ERC) under the European Union's Horizon 2020 research and inno-vation programme (grant agreement no. 755876 to T.M.). The experiment was performed in a controlled aquarium system which was funded by Institutional ISF grants 2288/16. Computations presented in this work were performed on the Hive computer cluster at the University of Haifa, which is partly funded by ISF grant 2155/15 to T.M.

Acknowledgements. We would like to thank Maayan Neder for the help with collecting coral larvae from the wild. We also thank Jeana Drake for the valuable and critical comments on the manuscript. Lastly, we thank the Interuniversity Institute of Marine Sciences in Eilat for access to its infrastructure and services.

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
