## [Peer Review File · Proceedings of the Royal Society B: Biological Sciences]

Review History

RSPB-2021-0328.R0 (Original submission)

Review form: Reviewer 1 (Senjie Lin)

Recommendation

Accept with minor revision (please list in comments)

Scientific importance: Is the manuscript an original and important contribution to its field?

Good

General interest: Is the paper of sufficient general interest?

Good

Quality of the paper: Is the overall quality of the paper suitable?

Good

Is the length of the paper justified?

Yes

Should the paper be seen by a specialist statistical reviewer?

No

Do you have any concerns about statistical analyses in this paper? If so, please specify them explicitly in your report.

No

It is a condition of publication that authors make their supporting data, code and materials available - either as supplementary material or hosted in an external repository. Please rate, if applicable, the supporting data on the following criteria.

Is it accessible?

No

Is it clear?

N/A

Is it adequate?

N/A

Do you have any ethical concerns with this paper?

No

Comments to the Author

This study investigates the effect of acidification on the primary polyps of the stony coral *Stylophora pistillata* using pH 7.8 and 7.6 versus pH 8.2 as the control. A set of morpho-structural, physiological, as well as transcriptomic parameters were measured. The major findings included 1) there was a greater response to the pH change from 8.2 to 7.6 than other changes; 2) skeleton calcium deposition was reduced but coral tissue growth was elevated under acidification; 3) genes related to calcification were downregulated but those related to environmental sensing and signaling were upregulated; 4) symbiont density increased, along with upregulation of genes involved in carbon fixation and photosynthate translocation. The authors conclude that the coral larvae were, at least some, more resilient to acidification and those that survived expressed mechanisms to counteract the effect of lower calcification under the extreme low pH.

The work is reasonably well executed. The approach of combining physiological and molecular analyses, and the analyses being done for both host and symbionts, is a big plus. The data are properly collected and interpreted. It is a pity that the Methods sections have to be presented in such a brief manner, going to the supplementary material is cumbersome for readers. I really wish that some can be presented in the main paper.

I recommend acceptance after minor revision. I have only a few technical comments for consideration in the revision, which are shown below.

What were the natural pH and other conditions from which the coral colonies were collected from?

With 9 day exposure to the acidification conditions, the responses observed were short-term/acute responses. For those that died or could not settle (would eventually die) the effect is clear. For those that survived, despite the observed structural, physiological and molecular alterations, would you expect them to recover, partially or in full, to the "normal" status (i.e. as if in pH8.2) if they were to be in the acidified condition over the long run? Is there any reported data of such recovery?

Related to the above question, if the observed effect will carry through adulthood and impact reproduction? I think it is important to give some discussion on the question here and the above.

Fig. 3B: GO terms clustered based on the similarity of gene content: gene numbers or gene functions, or degree of enrichment of the pathway by DEGs? Can the dot size be variable to

reflect the degree of enrichment? What about the significance level of the enrichment?

Fig. 4: The same questions as for 3B.

Review form: Reviewer 2 (Ewelina Rubin)

Recommendation

Accept with minor revision (please list in comments)

Scientific importance: Is the manuscript an original and important contribution to its field?

Excellent

General interest: Is the paper of sufficient general interest?

Excellent

Quality of the paper: Is the overall quality of the paper suitable?

Excellent

Is the length of the paper justified?

Yes

Should the paper be seen by a specialist statistical reviewer?

No

Do you have any concerns about statistical analyses in this paper? If so, please specify them explicitly in your report.

No

It is a condition of publication that authors make their supporting data, code and materials available - either as supplementary material or hosted in an external repository. Please rate, if applicable, the supporting data on the following criteria.

Is it accessible?

Yes

Is it clear?

Yes

Is it adequate?

Yes

Do you have any ethical concerns with this paper?

No

Comments to the Author

I think you did fantastic work on the study and the paper. I think the whole coral research field can benefit from studies that combine to measure transcriptomic responses and phenotypic traits. We need more studies like that. Those actually do move the coral research forward. it was a true please to read your paper.

I only have a few comments/suggestions for corrections

1) to my knowledge you are not the first presenting both coral and algal symbiont response to

OA (please check

Lin, Z., Wang, L., Chen, M., and Chen, J. (2018). The acute transcriptomic response of coral-algae interactions to pH fluctuation. *Mar Genomics* 42, 32-40.

2) there are few times in the discussion when you are using abbreviations of genes/protein (e.g. CARP4 and STPC) that you never defined in the paper before; I had to go to our supplementary tables but even there the proteins are not defined so it forces a reader to seek out this information in the reference material

3) I would not call the genes/proteins in table S4 to be "known to be biomineralization related" -- I would call them putative associated with the biomineralization process. Because sometimes the "evidence" comes only from transcriptomics studies, only and not direct functional genomics studies. In my opinion, we can not be sure that something that was regulated is involved in biomineralization.

I see many proteins that are known to be associated with other biological processes. For example -- vitellogenin -- there is direct evidence that the protein is present in coral oocytes -- see Shikina, S., Chen, C.J., Chung, Y.J., Shao, Z.F., Liou, J.Y., Tseng, H.P., Lee, Y.H., and Chang, C.F. (2013). Yolk formation in a stony coral *Euphyllia ancora* (Cnidaria, Anthozoa): insight into the evolution of vitellogenesis in nonbilaterian animals. *Endocrinology* 154, 3447-3459.

Great job again.

Decision letter (RSPB-2021-0328.R0)

17-May-2021

Dear Mrs Scucchia:

Your manuscript has now been peer reviewed and the reviews have been assessed by an Associate Editor. The reviewers' comments (not including confidential comments to the Editor) and the comments from the Associate Editor are included at the end of this email for your reference. As you will see, the reviewers and the Editors have raised some concerns with your manuscript and we would like to invite you to revise your manuscript to address them.

Research ethics:

Use of animals and field studies:

It is a condition of publication that you make available the data and research materials supporting the results in the article. Please see our Data Sharing Policies (<https://royalsociety.org/journals/authors/author-guidelines/#data>). Datasets should be deposited in an appropriate publicly available repository and details of the associated accession number, link or DOI to the datasets must be included in the Data Accessibility section of the article (<https://royalsociety.org/journals/ethics-policies/data-sharing-mining/>). Reference(s) to datasets should also be included in the reference list of the article with DOIs (where available).

Please submit a copy of your revised paper within three weeks. If we do not hear from you within this time your manuscript will be rejected. If you are unable to meet this deadline please let us know as soon as possible, as we may be able to grant a short extension.

Best wishes,
Dr Daniel Costa
mailto: proceedingsb@royalsociety.org

Reviewer(s)' Comments to Author:

Referee: 1

Comments to the Author(s)

This study investigates the effect of acidification on the primary polyps of the stony coral *Stylophora pistillata* using pH 7.8 and 7.6 versus pH 8.2 as the control. A set of morpho-structural, physiological, as well as transcriptomic parameters were measured. The major findings included 1) there was a greater response to the pH change from 8.2 to 7.6 than other changes; 2) skeleton calcium deposition was reduced but coral tissue growth was elevated under acidification; 3) genes related to calcification were downregulated but those related to environmental sensing and signaling were upregulated; 4) symbiont density increased, along with upregulation of genes involved in carbon fixation and photosynthate translocation. The authors conclude that the coral larvae were, at least some, more resilient to acidification and those that survived expressed mechanisms to counteract the effect of lower calcification under the extreme low pH.

The work is reasonably well executed. The approach of combining physiological and molecular analyses, and the analyses being done for both host and symbionts, is a big plus. The data are properly collected and interpreted. It is a pity that the Methods sections have to be presented in such a brief manner, going to the supplementary material is cumbersome for readers. I really wish that some can be presented in the main paper.

I recommend acceptance after minor revision. I have only a few technical comments for consideration in the revision, which are shown below.

What were the natural pH and other conditions from which the coral colonies were collected from?

With 9 day exposure to the acidification conditions, the responses observed were short-term/acute responses. For those that died or could not settle (would eventually die) the effect is clear. For those that survived, despite the observed structural, physiological and molecular alterations, would you expect them to recover, partially or in full, to the "normal" status (i.e. as if in pH8.2) if they were to be in the acidified condition over the long run? Is there any reported data of such recovery?

Related to the above question, if the observed effect will carry through adulthood and impact reproduction? I think it is important to give some discussion on the question here and the above.

Fig. 3B: GO terms clustered based on the similarity of gene content: gene numbers or gene functions, or degree of enrichment of the pathway by DEGs? Can the dot size be variable to reflect the degree of enrichment? What about the significance level of the enrichment?

Fig. 4: The same questions as for 3B.

Referee: 2

Comments to the Author(s)

I think you did fantastic work on the study and the paper. I think the whole coral research field can benefit from studies that combine to measure transcriptomic responses and phenotypic traits. We need more studies like that. Those actually do move the coral research forward. it was a true please to read your paper.

I only have a few comments/suggestions for corrections

1) to my knowledge you are not the first presenting both coral and algal symbiont response to OA (please check

Lin, Z., Wang, L., Chen, M., and Chen, J. (2018). The acute transcriptomic response of coral-algae interactions to pH fluctuation. *Mar Genomics* 42, 32-40.

2) there are few times in the discussion when you are using abbreviations of genes/protein (e.g. CARP4 and STPC) that you never defined in the paper before; I had to go to our supplementary tables but even there the proteins are not defined so it forces a reader to seek out this information in the reference material

3) I would not call the genes/proteins in table S4 to be "known to be biomineralization related" -- I would call them putative associated with the biomineralization process. Because sometimes the "evidence" comes only from transcriptomics studies, only and not direct functional genomics studies. In my opinion, we can not be sure that something that was regulated is involved in biomineralization.

I see many proteins that are known to be associated with other biological processes. For example -- vitellogenin -- there is direct evidence that the protein is present in coral oocytes -- see Shikina, S., Chen, C.J., Chung, Y.J., Shao, Z.F., Liou, J.Y., Tseng, H.P., Lee, Y.H., and Chang, C.F. (2013). Yolk formation in a stony coral *Euphyllia ancora* (Cnidaria, Anthozoa): insight into the evolution of vitellogenesis in nonbilaterian animals. *Endocrinology* 154, 3447-3459.

Great job again.

Author's Response to Decision Letter for (RSPB-2021-0328.R0)

See Appendix A.

Decision letter (RSPB-2021-0328.R1)

02-Jun-2021

Dear Mrs Scucchia

I am pleased to inform you that your manuscript entitled "Combined responses of primary coral polyps and their algal endosymbionts to decreasing seawater pH" has been accepted for publication in *Proceedings B*.

You can expect to receive a proof of your article from our Production office in due course, please check your spam filter if you do not receive it. PLEASE NOTE: you will be given the exact page

length of your paper which may be different from the estimation from Editorial and you may be asked to reduce your paper if it goes over the 10 page limit.

Data Accessibility section

Open Access

Paper charges

Sincerely,

Dr Daniel Costa

Appendix A

Response to reviewer's comments: Combined responses of primary coral polyps and their algal endosymbionts to decreasing seawater pH

Scucchia Federica, Malik Assaf, Zaslansky Paul, Putnam Hollie, Mass Tali

Dear editor and reviewers,

your constructive and valuable comments helped us in significantly improving the quality of the manuscript. We have addressed all the comments by the reviewers as explained below in this document. Further, the modifications we implemented are all marked in both the manuscript and the supporting information file, for ease of identification.

Below, we reproduce the reviewers' comments and questions in normal font with our responses in blue. Text that we have changed in the manuscript and copied into our responses is *italics*.

My co-authors and I thank you very much for your time and support and hope you find this experimental work relevant for your readership.

Sincerely,
Federica Scucchia

Detailed response to reviewers

Reviewer 1

- The work is reasonably well executed. The approach of combining physiological and molecular analyses, and the analyses being done for both host and symbionts, is a big plus. The data are properly collected and interpreted. It is a pity that the Methods sections have to be presented in such a brief manner, going to the supplementary material is cumbersome for readers. I really wish that some can be presented in the main paper.

Thank you for the positive comments. We agree with the reviewer and acknowledge that placing the Methods sections in the supplementary could be unwieldy for readers. However, to be able to obey to the strict word limit requirement of the journal, we had to take the entire Methods section out of the main manuscript which, in the current form, is already very close to the word limit.

- What were the natural pH and other conditions from which the coral colonies were collected from?

The natural water conditions of the site from which coral larvae were collected are available through the Israel National Monitoring Program of the Gulf of Eilat

http://www.meteo-tech.co.il/EilatYam_data/ey_data.asp), which maintains a long-term scientific data base of the major parameters (ecological, physical and chemical) encompassing the marine ecosystems at the northern head of the Gulf of Eilat.

We have now added this information in the Supplementary Methods, together with an additional reference for the light intensity conditions of the sampling site:

-lines 41-45 *“replicating the spring northern Red Sea water conditions that are available through the Israel National Monitoring Program of the Gulf of Eilat (http://www.meteo-tech.co.il/EilatYam_data/ey_data.asp): pH of 8.19 ± 0.01 (seasonal mean \pm s.d.), salinity of 40.63 ± 0.03 g liter⁻¹ (seasonal mean \pm s.d.), temperature of 22.18 ± 0.17 °C (seasonal mean \pm s.d.)”; “An irradiance of 250 $\mu\text{mol photons m}^{-2} \text{s}^{-1}$ (light conditions similar to those measured at the sampling site (Banc-Prandi et al., 2021)) on a 12-hour/12-hour photoperiod was provided”.*

- With 9 day exposure to the acidification conditions, the responses observed were short-term/acute responses. For those that died or could not settle (would eventually die) the effect is clear. For those that survived, despite the observed structural, physiological and molecular alterations, would you expect them to recover, partially or in full, to the “normal” status (i.e. as if in pH 8.2) if they were to be in the acidified condition over the long run? Is there any reported data of such recovery?

We are not aware of any study that assessed the degree of recovery in subsequent coral developmental stages under acidification conditions. However, a study conducted on adult corals reported a successful recovery from decalcification after a long-term exposure to OA (12 months), when corals were transferred back to ambient pH conditions (Fine & Tchernov, 2007), showing that corals indeed possess a remarkable ability to adjust and survive to future acidification conditions.

With our work, we thus hope to stimulate more research focused on the degree of modifications occurring in subsequent coral life stages exposed to long-term acidification conditions, to allow us to make more accurate predictions on the vulnerability of corals to future OA scenarios.

We have now added a brief discussion on this question in the manuscript:

-lines 285-290 *“Similarly to our study, a higher-biomass phenotype was observed under a long-term exposure to OA in the temperate coral *Oculina patagonica* (30). When transferred back to ambient pH conditions, these corals returned to calcify normally despite being 12 months as soft-bodied polyps in low-pH conditions (30), showing that corals possess a remarkable ability to adjust and survive to future acidification conditions.*

-lines 385-389 *“and on the phenotypic changes that could occur at later coral life stages under long-term acidification exposure. Thus, future effort should focus on assessing the degree of modifications in subsequent developmental stages of various coral species under acidification conditions, allowing us to make more accurate predictions on the vulnerability of corals to future OA scenarios”.*

- Related to the above question, if the observed effect will carry through adulthood and impact reproduction? I think it is important to give some discussion on the question here and the above.

To the best of our knowledge, there are no studies investigating carry-over effects of the changes occurring in coral recruits due to acidification that have an impact on reproduction. There is however evidence of corals that had lived all their post-settlement lives at naturally low pH conditions (volcanic CO₂ seeps) that showed no difference in fecundity as compared to corals living in non-acidic waters (Fabricius et al., 2017). We also would like to point out that such corals at naturally low pH conditions also showed improved adult performance as compared to corals living in non-acidic waters, to potentially compensate for their low recruitment rates (Fabricius et al., 2017).

Moreover, it appears that the shift to a less-skeleton/more-tissue phenotype (which is what we observed in the recruits) would not negatively impact the achievement of sexual maturity and gametogenesis, which appeared to develop similarly in both reduced-skeleton adult corals at acidified conditions and corals at ambient pH (Fine & Tchernov, 2007).

We have now added a brief discussion on this question in the manuscript:

-lines 252-253 *“Moreover, improved adult performance could compensate for low recruitment rates, as observed for corals living around CO₂ seeps (Fabricius et al., 2017)”*;

-lines 289-292 *“The shift to a less-skeleton/more-tissue phenotype would not negatively impact the achievement of sexual maturity and gametogenesis, which appeared to develop similarly in both reduced-skeleton corals at acidified conditions and corals at ambient pH (Fine & Tchernov, 2007)”*.

- Fig. 3B: GO terms clustered based on the similarity of gene content: gene numbers or gene functions, or degree of enrichment of the pathway by DEGs? Can the dot size be variable to reflect the degree of enrichment? What about the significance level of the enrichment?

GO terms were clustered based on the portion of shared genes (i.e. tightly clustered terms share a higher number of identical genes). We have now better clarified this point in the figure description:

-line 571 *“Dendrogram on the left showing enriched GO terms and KEGG pathways clustered according to the portion of identical genes shared”*.

We have modified the figure and changed the squares size based on significance level. We have also added the total number of genes within each GO term or KEGG pathway, to provide a clearer indication of the degree of enrichment:

-line 574 *“Red numbers indicate the total number of genes within each per GO term or KEGG pathway”*.

- Fig. 4: The same questions as for 3B.

We have improved the figure by adding the same changes as in Fig. 3B and modifying the figure description:

-lines 590-591 *"Dendrogram showing enriched functional groups clustered according to the portion of identical genes shared"*;

-line 595 *"Red numbers indicate the total number of genes within each functional group"*.

Reviewer 2

- to my knowledge you are not the first presenting both coral and algal symbiont response to OA (please check Lin, Z., Wang, L., Chen, M., and Chen, J. (2018). The acute transcriptomic response of coral-algae interactions to pH fluctuation. *Mar Genomics* 42, 32-40.

We thank you the reviewer for the comment, and modified the text in the manuscript as follows:

-lines 355-360 *"Earlier works on adult corals investigating the response to OA of the algal endosymbiont focused either on transcriptomic (Lin et al., 2018) or physiological changes (Baghdasarian et al., 2017; Kaniewska et al., 2012), and reported negative or no physiological responses in the algal symbionts with the exposure to acidic seawater. However, it must be noticed that broad physiological differences exist among different species across Symbiodiniaceae (Díaz-Almeyda et al., 2017), and that gene expression changes of the endosymbiotic algae greatly vary among coral populations in response to acidification (Kenkel et al., 2018)"*.

- there are few times in the discussion when you are using abbreviations of genes/protein (e.g. CARP4 and STPC) that you never defined in the paper before; I had to go to our supplementary tables but even there the proteins are not defined so it forces a reader to seek out this information in the reference material.

We appreciate the reviewer's useful suggestion and have now added the full gene names in the manuscript text (lines 206, 210): *"coral acid-rich protein 4 (CARP4)"*; *"Stylopora pistillata carbonic anhydrase 2 (STPCA-2)"*.

- I would not call the genes/proteins in table S4 to be "known to be biomineralization related" -- I would call them putative associated with the biomineralization process. Because sometimes the "evidence" comes only from transcriptomics studies, only and not direct functional genomics studies. In my opinion, we can not be sure that something that was regulated is involved in biomineralization. I see many proteins that are known to be associated with other biological processes. For example -- vitellogenin -- there is direct evidence that the protein is present in coral oocytes -- see Shikina, S., Chen, C.J., Chung, Y.J., Shao, Z.F., Liou, J.Y., Tseng, H.P., Lee, Y.H., and Chang, C.F. (2013). *Yolk formation in a stony coral Euphyllia ancora (Cnidaria, Anthozoa):*

insight into the evolution of vitellogenesis in nonbilaterian animals. *Endocrinology* 154, 3447-3459.

We thank the reviewer for the valuable suggestion and have now changed the genes description as “*putative biomineralization-related genes*”, both in the manuscript text (lines 195, 197), and in the supplementary file (lines 237, 241, description of Table S4).

References

- Baghdasarian, G., Osberg, A., Mihora, D., Putnam, H., Gates, R. D., & Edmunds, P. J. (2017). Effects of Temperature and $p\text{CO}_2$ on Population Regulation of *Symbiodinium* spp. in a Tropical Reef Coral. *The Biological Bulletin*, 232(2), 123–139. <https://doi.org/10.1086/692718>
- Banc-Prandi, G., Cerutti, J. M. B., & Fine, M. (2021). Recovery assessment of the branching coral *Stylophora pistillata* following copper contamination and depuration. *Marine Pollution Bulletin*, 162, 111830. <https://doi.org/10.1016/j.marpolbul.2020.111830>
- Díaz-Almeyda, E. M., Prada, C., Ohdera, A. H., Moran, H., Civitello, D. J., Iglesias-Prieto, R., Carlo, T. A., LaJeunesse, T. C., & Medina, M. (2017). Intraspecific and interspecific variation in thermotolerance and photoacclimation in *Symbiodinium* dinoflagellates. *Proceedings of the Royal Society B: Biological Sciences*, 284(1868), 20171767. <https://doi.org/10.1098/rspb.2017.1767>
- Fabricius, K. E., Noonan, S. H. C., Abrego, D., Harrington, L., & De'ath, G. (2017). Low recruitment due to altered settlement substrata as primary constraint for coral communities under ocean acidification. *Proceedings of the Royal Society B: Biological Sciences*, 284(1862), 20171536. <https://doi.org/10.1098/rspb.2017.1536>
- Fine, M., & Tchernov, D. (2007). Scleractinian Coral Species Survive and Recover from Decalcification. *Science*, 315(5820), 1811–1811. <https://doi.org/10.1126/science.1137094>
- Kaniewska, P., Campbell, P. R., Kline, D. I., Rodriguez-Lanetty, M., Miller, D. J., Dove, S., & Hoegh-Guldberg, O. (2012). Major Cellular and Physiological Impacts of Ocean Acidification on a Reef Building Coral. *PLoS ONE*, 7(4), e34659. <https://doi.org/10.1371/journal.pone.0034659>
- Kenkel, C. D., Moya, A., Strahl, J., Humphrey, C., & Bay, L. K. (2018). Functional genomic analysis of corals from natural CO_2 -seeps reveals core molecular responses involved in acclimatization to ocean acidification. *Global Change Biology*, 24(1), 158–171. <https://doi.org/10.1111/gcb.13833>
- Lin, Z., Wang, L., Chen, M., & Chen, J. (2018). The acute transcriptomic response of coral-algae interactions to pH fluctuation. *Marine Genomics*, 42, 32–40. <https://doi.org/10.1016/j.margen.2018.08.006>